# Testing for Vitamin D in High-Risk COPD in Outpatient Clinics in Spain: A Cross-Sectional Analysis of the VITADEPOC Study

**DOI:** 10.3390/jcm11051347

**Published:** 2022-03-01

**Authors:** Myriam Calle Rubio, José Luis Álvarez-Sala, Gianna Vargas Centanaro, Ana María Humanes Navarro, Juan Luis Rodríguez Hermosa

**Affiliations:** 1Pulmonary Department, Research Institute of Hospital Clínico San Carlos (IdISSC), 28040 Madrid, Spain; mcal01@ucm.es (M.C.R.); jalvar01@ucm.es (J.L.Á.-S.); gvargas@alumni.unav.es (G.V.C.); 2Department of Medicine, Faculty of Medicine, University Complutense of Madrid, 28040 Madrid, Spain; 3Clinical Management Unit of Medicina Preventiva, Research Institute of Hospital Clínico San Carlos (IdISSC), 28040 Madrid, Spain; anahumnav@gmail.com

**Keywords:** vitamin D deficiency, testing for vitamin D, chronic obstructive pulmonary disease (COPD), high risk, adherence to recommendation

## Abstract

Background: Vitamin D deficiency has been associated with an accelerated deterioration in lung function and increased exacerbations in chronic obstructive pulmonary disease (COPD). 25(OH) vitamin D levels have been indicated as a potentially useful marker for adverse results related to COPD. Methods: VITADEPOC is a cross-sectional clinical study recruiting consecutive patients with high-risk COPD. The objective of our study was to investigate vitamin D determination frequency in patients with high-risk COPD in clinical practice at outpatient clinics in Spain and to describe the factors associated with vitamin D testing. We also aimed to determine the frequency of vitamin D deficiency in these patients. Results: Only 51 (44%) patients underwent vitamin D determination and 33 (28.4%) had received vitamin D supplements in clinical practice. The patients who underwent testing for vitamin D in clinical practice were more often women (58.8% vs. 26.2%, *p* < 0.001) with comorbidities such as osteoporosis (19.6% vs. 6.2%, *p* < 0.001) or chronic renal failure (7.8% vs. 0%, *p* < 0.001) and with exacerbator phenotype (55% vs. 32.3%, *p* = 0.015). A total of 63 (54.3%) patients had serum vitamin D levels <20 ng/mL at the inclusion visit. Of these, 29 (46%) had serum vitamin D levels <12 ng/mL (severe deficiency). Having a history of inhaled corticosteroids (OR 3.210, *p* < 0.016), being treated with a cycle of systemic corticosteroids (OR 2.149, *p* < 0.002), and having a lower physical activity level (OR 3.840, *p* < 0.004) showed a statistically significant positive association with vitamin D deficiency. Conclusion: The testing of vitamin D levels in patients with high-risk COPD treated at outpatient respiratory clinics in Spain is infrequent. However, when tested, a severe deficiency is detected in one in four patients. Efforts to optimize case detection in COPD are needed.

## 1. Introduction

Chronic pulmonary obstructive disease (COPD) places an enormous burden on healthcare systems. A substantial proportion of the cost is attributable to hospitalizations, mostly due to acute exacerbations of respiratory symptoms [1]. COPD exacerbations have important consequences for patients and healthcare providers; they cause a negative impact on health-related quality of life [2], a decline in pulmonary function [3], increased utilization of healthcare resources [4], and decreased survival [5,6].

In recent years, the approach to COPD has changed to offer more personalized medicine [7]. COPD is a complex disease that requires risk assessment in order to offer high-risk patients better diagnostic precision and a more personalized treatment plan.

Patients with frequent exacerbations are specifically targeted with more aggressive therapy and an action plan in order to help prevent exacerbations [8,9].

Vitamin D deficiency is common and represents a major health problem. Epidemiological studies have shown low vitamin D nutritional status in highly prevalent chronic diseases, including autoimmune disease, infectious disease, allergic disease, endocrine and metabolic disorders, cancer, cardiovascular disease, and COPD [10,11,12].

Vitamin D is traditionally known for its roles in bone health and the homeostasis of calcium and phosphorus. In adults, vitamin D deficiency is known to accelerate osteopenia and osteoporosis [13]. Studies have also suggested a far broader range of physiological effects of vitamin D, including effects on muscle function and the immune system, which are linked to the presence of vitamin D receptors (VDR) on the cells of these tissues [14]. The immuno-modulating role is well known, as is its ability to promote innate immune responses to infections, as well as to regulate the adaptive immune response, along with the antimicrobial and anti-inflammatory activity of active vitamin D metabolites shown in “in vitro” studies at the lung level [15,16]. In addition to this, studies have shown an association between respiratory infections and low vitamin D levels [17]. These effects might have clinical implications for patients with COPD, especially in some patients who are more susceptible to developing exacerbations, termed frequent exacerbators or COPD exacerbator phenotype [18].

Numerous studies have shown us that vitamin D deficiency is associated with worse lung function, an accelerated deterioration in lung function, and an increase in COPD exacerbations [19]. These findings identify 25(OH) vitamin D levels as a potentially useful marker for the adverse results related to COPD [20]. Clinical trials have also shown the benefits of cholecalciferol supplements in patients with COPD in clinical terms as improved FEV1 and quality of life and reduction in exacerbation rates [21]. Supplementing this vitamin in patients with severe deficit (10 ng/mL or 25 nmol/L) has been shown to reduce the rate of exacerbations in moderate/severe COPD (CI 95%: 0.36 to 0.84) [21,22]. These findings are the basis for the current recommendations for good clinical practice, which establish that in patients with high risk and clinical impact, especially if they are frequent exacerbators, 25(OH) vitamin D serum concentrations should be determined and supplements taken if 25(OH) vitamin D levels are below 10 ng/mL or 25 nmol/L [23].

The objective of our study was to investigate the frequency of vitamin D determination in patients with high-risk COPD in clinical practice at outpatient clinics in Spain and to describe the factors associated with vitamin D testing. We also aimed to determine the frequency of vitamin D deficiency in these patients and to describe their clinical characteristics.

## 2. Materials and Methods

A transversal observational study was carried out between November 2019 and February 2020. The sample population was patients treated at primary care centers and the pulmonology clinic for the population treated at the Hospital Clínico San Carlos, Madrid, Spain. Recruitment was prospective, and the first ten patients with a diagnosis of COPD who consecutively came to the clinic for a checkup were included. The inclusion criteria were patients aged >40 years, smokers or ex-smokers (of at least 10 pack-years) with COPD diagnosed on the basis of spirometry tests (FEV1/FVC post-bronchodilation <0.7 or FEV1/FVC pre-bronchodilation <0.7 + FEV1 ≤80% if there was no bronchodilation reversibility testing available), and who were at high risk according to GesEPOC criteria [8]. The exclusion criteria were undergoing no previous follow-up in an outpatient clinic for at least 1 year or the inability to fill out the quality of life questionnaires or participate in a clinical study.

The information collected was concurrent for clinical data (during the only visit) and historical for the evaluation of clinical data and tests conducted based on the visit history recorded during previous visits. Telephonic follow-up with the participating researchers was carried out in order to guarantee the scientific and methodological rigor of the study.

### Variable Selection

The data collected for patients concerned sociodemographic and clinical COPD characteristics. The level of dyspnea was evaluated according to the mMRC (Modified Medical Research Council) dyspnea scale [24]. Comorbidities were evaluated using the Charlson index [25], and physical activity level was assessed with the IPAQ questionnaire [26]. Quality of life was assessed using the CAT (COPD Assessment Test) [27]. The exacerbator phenotype was defined according to GesEPOC criteria [28]: a patient who had at least two exacerbations or one hospitalization in the previous year.

The level of sun exposure was established according to the following criteria: high exposure (referring to 3 or more hours of exposure per day, for at least five days a week, in the past three months), medium exposure (referring to 1 to 2 h of exposure per day, for at least five days a week, in the past three months), and low exposure (referring to less than 1 h of exposure per day, less than five days a week, in the past three months).

Serum 25(OH) vitamin D was measured using the liquid chromatography coupled to tandem mass spectrometry method. All 25(OH) vitamin D data in this study represent a single time point, measured in patients with stable COPD. Vitamin D deficiency was defined as a 25(OH) vitamin D level less than 20 ng/mL [29,30].

Qualitative variables are presented with their frequency distribution, and quantitative variables are summarized as average and standard deviation (SD). The quantitative variables that showed an asymmetrical distribution are summarized as median and interquartile range (IQR).

To compare qualitative variables, the chi-square test or Fisher’s exact test was used, as necessary. Averages between two independent groups were compared using Student’s t test, if the variables followed a normal distribution, or using the nonparametric Mann–Whitney U test for asymmetrical variables. Averages between more than two independent groups were compared using the analysis of variance (ANOVA) or the nonparametric Kruskal–Wallis test for asymmetric variables.

The association between each independent variable (patient characteristics) and the dependent variable (vitamin D level </> = 20 ng/dL) was evaluated by calculating the odds ratio (OR) using a logistic regression model. Discrimination was quantified by calculating the area under the curve (AUC) for the ROC curve and its respective confidence interval (CI).

For the linear association study (correlation) between vitamin D levels and the continuous independent variables in the study, Spearman’s nonparametric correlation coefficient was calculated.

A level of significance of 5% was accepted for all tests. Data processing and analysis were carried out using IBM SPSS Statistics v.26 software.

## 3. Results

### 3.1. Population

A total of 116 patients was included in the analysis. The sampling process, detailed in an epidemiology flow chart, is presented in Figure 1.

### 3.2. Vitamin D Testing in Patients with High-Risk COPD

Of the analyzed cohort, only 51 (44%) patients underwent vitamin D determination, and 33 (28.4%) had received vitamin D supplements in clinical practice.

The mean vitamin D level when the test was performed was 11.9 (8.6–17.5) ng/mL. In total, 26 patients (51% of the individuals tested) had serum vitamin D levels <20 ng/mL. Of these, 15 (29.4%) patients had serum vitamin D levels <12 ng/mL (severe deficiency).

Figure 2 describes the sociodemographic and clinical characteristics of patients with high-risk COPD and the association with vitamin D testing in clinical practice (univariate logistic regression analysis). Few patient-level variables were associated with vitamin D level determination. The patients who underwent testing for vitamin D in clinical practice were more often women (58.8% vs. 26.2%, *p* < 0.001), were more likely to have predisposing comorbidities such as osteoporosis (19.6% vs. 6.2%, *p* < 0.001) or chronic renal failure (7.8% vs. 0%, *p* < 0.001), and were more likely to have the exacerbator phenotype (55% vs. 32.3%, *p* = 0.015).

In total, 33 patients (64.7% of the individuals tested) had received vitamin D supplements in clinical practice. Table 1 describes the characteristics of the patients with high-risk COPD and the association with receiving vitamin D supplements in clinical practice (univariate logistic regression analysis). Patients who had received vitamin D supplementation had a lower vitamin D level (11.9 (8.6–17.5) vs. 17.3 (14.3–21.8), *p* = 0.003)), and a history of hospitalization due to COPD exacerbation was more frequent (35.3% vs. 21.2%, *p* = 0.004).

### 3.3. Vitamin D Deficiency and High-Risk COPD

Sixty-three (54.3%) patients had serum vitamin D levels <20 ng/mL. Of these, 29 (46%) had serum vitamin D levels <12 ng/mL (severe deficiency). Patients with vitamin D deficiency (based on serum vitamin D levels) had lower physical activity levels, MET (422 (242.5–1181) vs. 1740 (714.5–2772), *p* < 0.001), had higher obstruction severity according to FEV1 (50.6 (21.3) vs. 60.7 (17.9), *p* = 0.007), and a greater impact on their quality of life as evaluated with the CAT (16 (9–22) vs. 10 (6.5–18.5), *p* = 0.033). They also had more hospitalizations (50.8% vs. 3.8%, *p* < 0.001) and were more likely to have the exacerbator phenotype (58.7% vs. 24.5%, *p* < 0.001).

In relation to the therapeutic management in clinical practice on record, oxygen therapy (50.8% vs. 28.3%, *p* = 0.014) and systemic corticosteroid therapy in the previous 6 months (55.6% vs. 17%, *p* = 0.003) were more frequently used in patients with vitamin D deficiency. At their previous visits, there were no significant vitamin D determination differences between patients with or without vitamin D deficiency at the inclusion visit (41.3% vs. 47.2%, *p* = 0.576). However, there were significant differences in having undergone treatment with vitamin D supplements, which was more frequent in patients without vitamin D deficiency at the current visit (39.6% vs. 19%, *p* = 0.021), as summarized in Table 2.

Multilevel analysis of factors associated with vitamin D deficiency in patients with high-risk COPD

In the adjusted model, summarized in Table 3, maintenance treatment with inhaled corticosteroids (OR 3.210, *p* < 0.016), being treated with a cycle of systemic corticosteroids in the six months prior (OR 2.149, *p* < 0.002), and having a lower (OR 3.840, *p* < 0.004) showed a statistically significant positive association with vitamin D deficiency. The relationship between physical activity level and cycles of systemic corticosteroids and vitamin D levels in COPD are described in Figure 3 and Figure 4.

## 4. Discussion

This study provides data for the first time about the frequency with which vitamin D levels in blood serum are determined in clinical practice in patients with high-risk COPD who are treated in outpatient clinics in Spain, as well as the factors associated with this determination.

Vitamin D deficiency has been shown to be very common among patients with COPD [19,31], and studies have shown that vitamin D deficiency is associated with worse lung function [32], an accelerated decrease in lung function, and an increase in COPD exacerbations [19]. Despite this evidence, the results of our study show that the majority of patients with high-risk COPD do not undergo vitamin D serum determination, despite having prolonged follow-up. These data reflect a clinical practice that does not adhere to the current good clinical practice guides for COPD with regard to the detection of cases of severe deficiency in an at-risk group, such as patients who have frequent exacerbations with a high clinical impact [23].

This is surprising, since most patients with COPD in our study are at an increased risk of having low vitamin D blood levels due to the presence of risk factors where vitamin D deficiency monitoring is recommended [30]. These risk factors are reduced outdoor activity, old age, smoking, and treatment with glucocorticoids.

The factors associated with serum vitamin D determination found in our study were clinical characteristics traditionally related to vitamin D deficiency. Therefore, patients who had undergone vitamin D determination were more frequently women with a history of osteoporosis or chronic renal failure. However, it must be noted that other risk factors for suffering vitamin D deficiency and which are frequently present in patients with COPD were not considered, such as active tobacco use, physical inactivity, and treatment with corticosteroids in the months prior.

In Spain, a study analyzing the number of requests for vitamin D determination in a given period (2010 to 2014) revealed figures of 1.1 per 1000 inhabitants in 2012 and 3.4 per 1000 inhabitants in 2014, with significant variability between different regional communities, ranging from 0.94 to 21.4 [33]. The number of requests was lower in areas with more hours of sunlight and in those areas with restrictive request criteria. These results show that although vitamin D determination seems frequent in Spain, according to our results, it does not seem to adhere to the current recommendations for risk groups. In this sense, some studies have shown that tools to aid decision-making incorporated in the digitized medical history can help align vitamin D determination in primary care [34]. There is currently no evidence that shows the benefits of vitamin D deficit screening in the general population [35,36]. The current recommendations developed by a consensus of experts advise determining serum 25(OH) vitamin D concentrations in patients with a higher risk of having vitamin D deficiency, such as those with renal or liver failure, hypo- or hyperparathyroidism, rickets, and osteoporosis or fragility fractures, as well as in patients treated with drugs that affect vitamin D absorption and/or metabolism, such as anticonvulsants, glucocorticoids, or antiretrovirals [30]. Other risk factors associated with hypovitaminosis and which will be present in patients with chronic diseases such as COPD should also be considered, such as old age, institutionalized individuals, those with cognitive impairment, central obesity, smokers, corticosteroid therapy, and those with limited sun exposure.

We must remember that cholecalciferol or “vitamin” D3 is synthesized by 7-dehydrocholestrol in the skin through ultraviolet B radiation (UVB). The main source of vitamin D is sun exposure between 10 am and 3 pm in the spring, summer, and fall. There is little to no vitamin D synthesis in the winter. This route accounts for around 80–90% of vitamin D in the body, with the remaining approximately 10–20% being obtained through diet [37]. There are few foods that naturally contain vitamin D in sufficient amounts to cover our daily needs. That is why the leading cause of vitamin D deficit is insufficient sun exposure, followed by the coexistence of clinical situations that limit vitamin D absorption or favor its loss.

As far as the prevalence of vitamin D deficiency in high-risk COPD, in our study, the detection rate was 63 (54.3%) deficient cases (25(OH) vitamin D <20 ng/mL) and 29 (25%) cases with severe deficit (25(OH) vitamin D <12 ng/mL). Studies have assessed vitamin D status in patients with COPD, reporting a highly variable prevalence, between 31% and 60%, but much greater when compared with a control group [19,29,31]. However, there are also not enough data to determine whether the results from these studies are applicable to specific subgroups of patients, such as patients with high-risk COPD. Thus, in a study conducted in patients with COPD who were hospitalized for exacerbations, the reported prevalence was 83.6% [38], and as many as 77% of GOLD stage IV patients were deficient in vitamin D [19].

In our study, it is important to note that the population evaluated included patients who met the criteria for high-risk-developed COPD, with severe airflow obstruction, and moderate dyspnea with minimal exertion, which resulted in the vast majority reporting low physical activity and sun exposure. In addition, almost half of the population evaluated were frequent exacerbators and required home oxygen therapy.

One of the main factors related to the presence of vitamin D deficiency in our population was having a low level of physical activity (3.8 times more likely), being treated with a cycle of systemic corticosteroids in the previous six months (2.1 times more likely), and maintenance therapy with inhaled corticosteroids (3.2 times more likely). We know that glucocorticoids decrease intestinal absorption and increase the urinary excretion of calcium, with the resulting increase in risk of osteoporosis and bone fracture [39]. However, beyond that, they also increase the activity of the enzyme 24-hydroxilase, which results in a decrease in vitamin D hormone production [40]. The NHANES (National Health and Nutrition Examination Survey) study [41] showed that the use of corticosteroids is independently associated with vitamin D deficiency, and thus exogenous vitamin D supplementation is recommended in cases of deficiency in patients undergoing chronic steroid treatment, as in the case of patients with COPD.

However, there were no differences in age, body mass index, the presence of predisposing comorbidities, current tobacco use, or obstruction severity. Upon analyzing the actions carried out in COPD follow-up, it is important to note that patients with vitamin D deficiency were less likely to undergo vitamin D determination or have a history of having received vitamin D supplements. With regard to supplementation and its monitoring, it is important to mention that the dose and frequency required will depend on the severity of the deficit, its causes, and also the used vitamin D form [30]. In our study, there was a limited number of patients with high-risk COPD who had received vitamin D supplements, despite the beneficial effects of vitamin D in reducing the risk of exacerbation [42,43]. In our analysis, the factors related to a history of vitamin D supplementation were having lower vitamin D levels and a history of hospitalization.

The main strength of this study is that it investigated the frequency of determining vitamin D in patients with high-risk COPD in clinical practice, and it describes the frequency of vitamin D deficiency in these patients at outpatient clinics in Spain. Another strength of our study is the fact that the inclusion period for all patients was carried out in the autumn and winter months, avoiding that seasonal variation that could be a biasing factor. In addition, all vitamin D determinations were made in the same laboratory, avoiding interlaboratory variations in vitamin D measurements that can contribute to the heterogeneity of vitamin D deficiency.

Some limitations are that this was a cross-sectional study, and the study sample size was small. The nature of the study design resulted in the identification of association but not causality. Additionally, the number of participating patients was relatively limited, with 116 patients evaluated at a single center. However, despite these limitations, we believe that the sample included is representative of medical attention for patients with high-risk COPD in outpatient clinics in Spain.

## 5. Conclusions

Vitamin D determination is carried out infrequently in patients with high-risk COPD in follow-up visits. The factor associated with vitamin D determination is the presence of clinical characteristics typically related to vitamin D deficiency’s effects on bone metabolism, which suggests that a selective search strategy is not followed when there are risk factors present, such as old age, the use of systemic corticosteroids, or limited sun exposure, which are frequent in this population.

Testing for vitamin D deficiency in patients with COPD and frequent exacerbations with a high clinical impact and treating the vitamin D deficiency with vitamin D supplements are important. As a result, we believe it is necessary to establish training and awareness programs for healthcare professionals and to use electronic medical registries in habitual clinical practice as support tools to avoid variability and improve care quality.

## Figures and Tables

**Figure 1 jcm-11-01347-f001:**
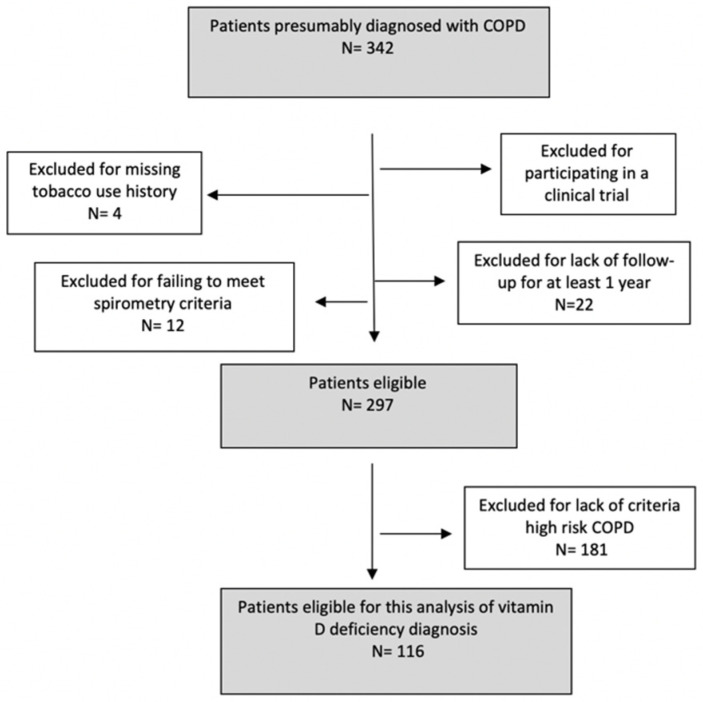
Sampling process as described in a STROBE flow chart.

**Figure 2 jcm-11-01347-f002:**
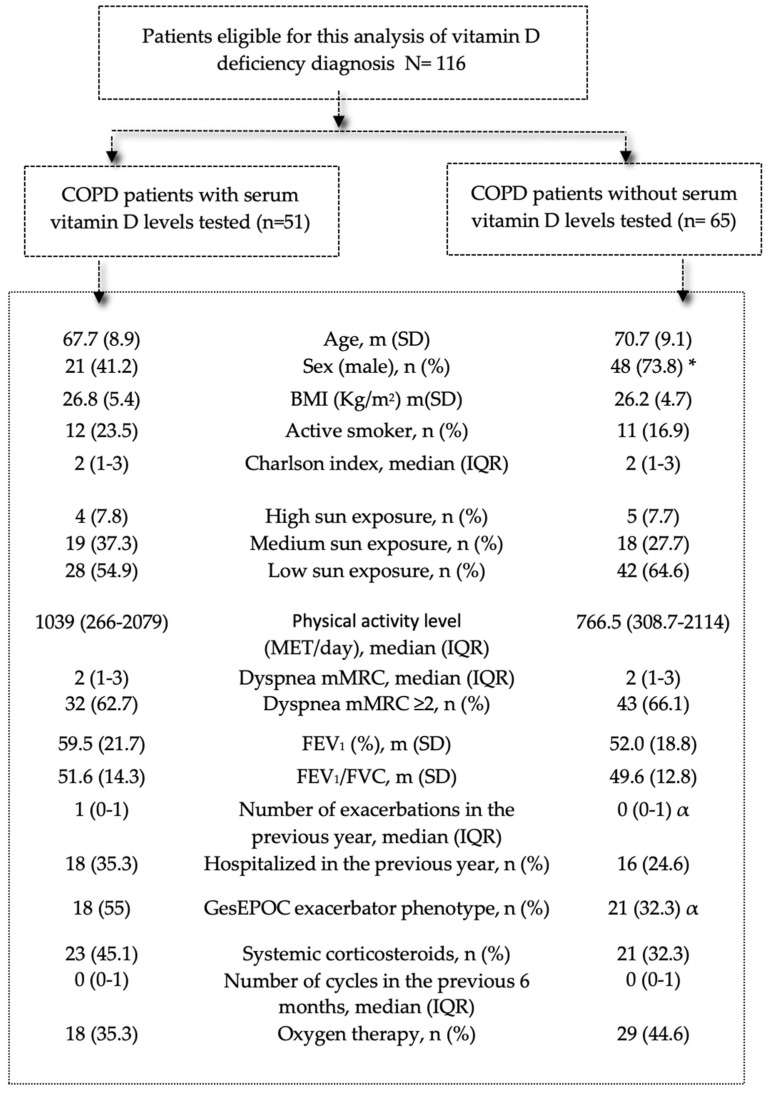
Characteristics of patients with COPD according to presence/absence of serum vitamin D level testing to detect vitamin D deficiency during follow-up in clinical practice. Note: Data are presented as mean (SD), number (%), or median (IQR). Abbreviations: IQR: interquartile range; BMI: body mass index; mMRC: modified Medical Research Council; MET: metabolic equivalent of task; FEV1: forced expiratory volume in 1 s; GesEPOC: Spanish National Guideline for COPD. * *p* < 0.001; α *p* < 0.05.

**Figure 3 jcm-11-01347-f003:**
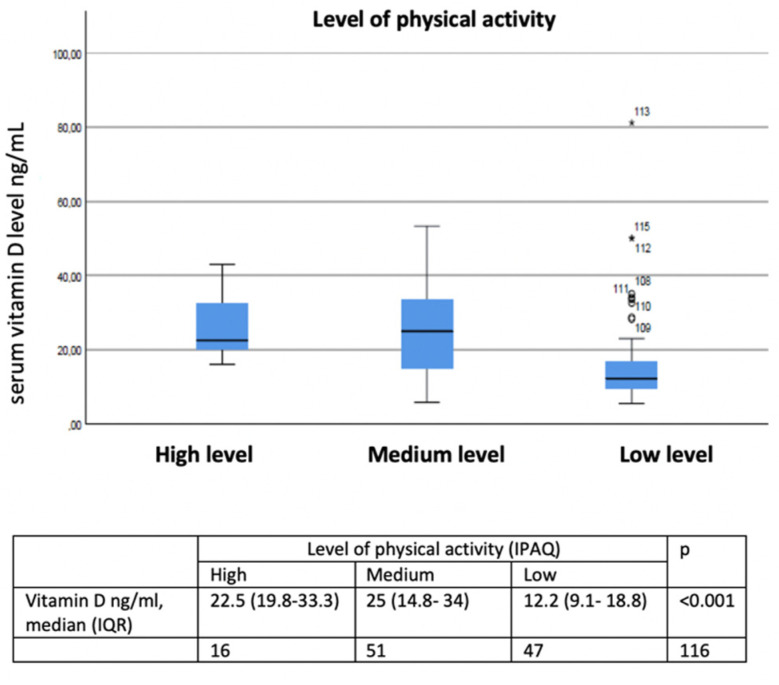
Relationship between low, medium, and high activity levels and vitamin D levels in COPD. ○ Outliers values: values more than 1.5 box lengths (IQR) away from the 25th and 75th percentile. * Extreme values: values more than 3 box lengths (IQR) away from the 25th and 75th percentile.

**Figure 4 jcm-11-01347-f004:**
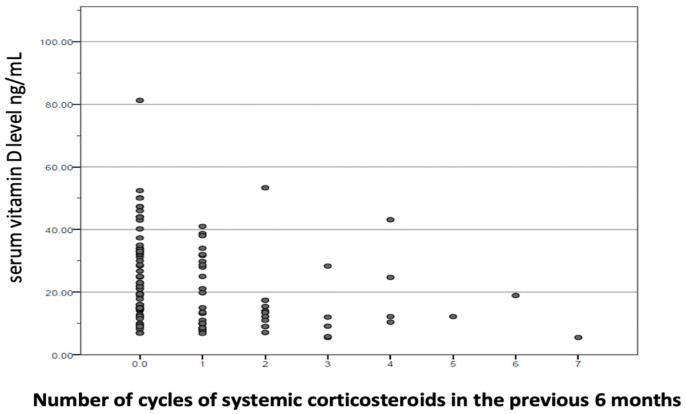
Relationship between number of cycles of systemic corticosteroids in the previous 6 months and vitamin D levels in COPD. ● Each dot on the scatterplot represents one observation from a data set. The position of the dot on the scatterplot represents its X and Y values.

**Table 1 jcm-11-01347-t001:** Characteristics of patients with COPD according to having received vitamin D supplementation in clinical practice.

Diagnosis Prior to Vitamin D	All	Received Vitamin D Supplementation	Did Not Receive Vitamin D Supplementation	*p*
	(*n* = 51)	(*n* = 33)	(*n* = 18)	
Age, m (SD)	67.7 (8.9)	66.9 (8.5)	69.3 (9.8)	0.368
Sex (male), *n* (%)	21 (41.2)	9 (27.3)	12 (66.7)	0.006
BMI (Kg/m^2^), m (SD)	26.8 (5.4)	27.1 (5.9)	26.4 (4.6)	0.666
Pack-years, median (IQR)	40 (24–60)	49 (21–71)	40 (28.5–50)	0.185
Active smoker, *n* (%)	12 (23.5)	8 (24.2)	4 (22.2)	1
Charlson index, median (IQR)	2 (1–3)	2 (1–3)	2 (2–3)	0.199
Predisposing comorbidities, *n* (%)	10 (19.6)	8 (24.2)	2 (11.1)	0.462
- Osteoporosis - Chronic renal failure	4 (7.8)	1 (3)	3 (16.7)	0.12

Sun exposure	4 (7.8)	2 (6.1)	2 (11.1)	0.537
- High exposure, *n* (%) - Medium exposure, *n* (%) - Low exposure, *n* (%)	19 (37.3)	14 (42.4)	5 (27.8)	
	28 (54.9)	17 (51.5)	11 (61.1)	

Physical activity level	1039 (266–2079)	1386 (288–2079)	656.5 (256.5–1559)	0.413
(MET), median (IQR)				
Physical activity level	7 (13.7)	5 (15.2)	2 (11.1)	0.476
- High, *n* (%) - Medium, *n* (%) - Low, *n* (%)	22 (43.1)	15 (45.5)	7 (38.9)	
	22 (43.1)	13 (39.4)	9 (50)	

Previous vitamin D level, median (IQR)	13.4 (8.5–17.8)	11.9 (8.6–17.5)	17.3 (14.3–21.8)	0.003
Dyspnea mMRC, median (IQR)	2 (1–3)	2 (1–3)	2 (1–3)	0.766
≥2, *n* (%)	32 (62.7)	20 (60.6)	12 (66.7)	0.767
CAT, m (SD)	14.1 (9.0)	13.5 (8.4)	15.2 (10.1)	0.664
CAT ≥10, *n* (%)	31 (60.8)	20 (60.6)	11 (61.1)	1
FEV_1_ (mL), median (IQR)	1290 (1090–1860)	1230 (1040–1780)	1525 (1047–1927)	0.364
FEV_1_ (%), m (SD)	59.5 (21.7)	60.7 (21.2)	57.2 (23.1)	0.597
FVC ml, median (IQR)	2550 (2130–3060)	2420 (2105–2920)	2860 (2197–3517)	0.104
FVC %, m (SD)	92.0 (19.5)	93.7 (18.9)	88.8 (20.7)	0.397
FEV_1_/FVC, m (SD)	51.6 (14.3)	52.4 (14.4)	50.2 (14.3)	0.597
Number of exacerbations in previous year, m (SD)	1 (0–1)	1 (1.7)	1.1 (1.2)	0.272
Hospitalizations	18 (35.3)	7 (21.2)	11 (61.1)	0.004
GesEPOC exacerbator phenotype, *n* (%)	18 (55)	17 (51.5)	11 (61.1)	0.548
Treatment, *n* (%)	3 (5.9)	1 (3)	1 (5.6)	0.344
- LAMA - LAMA + LABA LABA + ICS - LAMA + LABA + ICS	28 (54.9)	15 (45.5)	13 (72.2)	
	1 (2)	1 (3)	0	
	19 (37.3)	15 845.5)	4 (22.2)	

Systemic corticosteroids, *n* (%)	23 (45.1)	12 (36.4)	11 (61.1)	0.141
Number of cycles, median (IQR)	0 (0–1)	0 (0–1)	1 (0–2)	0.061
Oxygen therapy, *n* (%)	18 (35.3)	12 (36.4)	6 (33.3)	1

Note: Data are presented as mean (SD), number (%), or median (IQR). Abbreviations: IQR: interquartile range; BMI: body mass index; mMRC: modified Medical Research Council; MET: metabolic equivalent of task; CAT: COPD Assessment Test; FEV1: forced expiratory volume in 1 s; FVC: forced vital capacity; GesEPOC: Spanish National Guideline for COPD; LABA: long-acting beta-2 agonists; LAMA: long-acting antimuscarinic agents; ICS: inhaled corticosteroids.

**Table 2 jcm-11-01347-t002:** Characteristics of patients with COPD according to presence/absence of vitamin D deficiency during visit.

	All	Vitamin D Levels <20 ng/mL	Vitamin D Levels ≥20 ng/mL	*p*
	(*n*= 116)	(*n* = 63)	(*n*= 53)	
Male, *n* (%)	69 (59.5)	40 (63.5)	29 (54.7)	0.338
Age, m (SD)	69.4 (9.1)	70.2 (8.6)	68.4 (68.5)	0.316
BMI (kg/m^2^), m (SD)	26.5 (5.0)	25.9 (4.5)	27.1 (5.5)	0.202
Pack-years, median (IQR)	45 (30–60)	50 (34–65)	42 (30–50)	0.078
Employment status				0.909
Active	12 (10.3)	7 (11.1)	5 (9.4)	
Retired	78 (67.2)	43 (68.3)	35 (66)	
Low/inability to work	21 (18.1)	10 (15.9)	11 (20.8)	
Unemployed	5 (4.3)	3 (4.8)	2 (3.8)	
Level of studies				0.059
High	27 (23.3)	11 (17.5)	16 (30.2)	
Average	53 (45.7)	35 (55.6)	18 (34)	
No studies	36 (31)	17 (27)	19 (35.8)	
Living situation				0.353
Single	30 (25.9)	14 (22.2)	16 (30.2)	
Partner	85 (73.3)	48 (76.2)	37 (69.8)	
Residence	1 (0.9)	1 (1.6)	0	
Charlson index, median (IQR)	2 (1- 3)	2 (1–3)	2 (1–3)	0.677
Predisposing comorbidities, *n* (%)				
Osteoporosis Chronic renal failure	14 (12.1)	2 (1–3)	2 (1–3)	0.667
	4 (3.4)	8 (12.7)	6 (11.3)	0.821
Active smoker, *n* (%)	23 (19.8)	9 (14.3)	14 (26.4)	0.103
Vitamin D levels al visit, m (SD)	18.8 (11.6–32)	12.11 (3.92)	34.54 (11.08)	<0.001
Vitamin D >30, *n* (%)	32 (27.6)		32 (60.3)	
Vitamin D ≤30 and ≥20, *n* (%)	21 (18.1)		21 (39.6)	
Vitamin D <20 and ≥12, *n* (%)	34 (29.3)	34 (54)		
Vitamin D <12, *n* (%)	29 (25)	29 (46)		
Physical activity level (MET), median (IQR)	880 (288–2079)	422 (242.5–1181)	1740 (714.5–2772)	<0.001
Physical activity level				
High, *n* (%) Medium, *n* (%) Low, *n* (%)	16 (13.8)	4 (6.6)	12 (22.6)	<0.001
	51 (43.9)	21 (34.4)	30 (56.6)	
	47 (40.5)	36 (59)	11 (20.8)	
Sun exposure				0.253
High exposure, *n* (%) Medium exposure, *n* (%) Low exposure, *n* (%)	9 (7.8)	5 (7.9)	4 (7.5)	
	37 (31.9)	16 (25.4)	21 (39.7)	
	70 (60.3)	42 (66.7)	28 (52.8)	
FEV1 mL, median (IQR)	1290 (957–1800)	1260 (900–1860)	1350 (1110–1730)	0.24
FEV1 (%), m (SD)	55.3 (20.4)	50.6 (21.3)	60.7 (17.9)	0.007
FVC ml, median (IQR)	2725 (2235–3325)	2790 (2210–3380)	2580 (2280–3320)	0.816
FVC %, m (SD)	90.6 (19.5)	89.6 (70.3–104.6)	92.2 (18.5)	0.421
FEV1/FVC, m (SD)	50.5 (13.5)	49 (39.3–60.8)	52.3 (12.6)	0.189
Dyspnea mMRC, median (IQR)	2 (1–3)	2 (1–3)	2 (1–2)	0.072
≥2, *n* (%)	75 (64.6)	45 (71.4)	30 (56.6)	0.096
CAT, median (IQR)	13.5 (7–20.7)	16 (9–22)	10 (6.5–18.5)	0.033
GesEPOC exacerbator phenotype, *n* (%)	56 (48.3)	37 (58.7)	13 (24.5)	<0.001
Number of exacerbations in previous year, median (IQR)	0 (0–1)	1 (0–2)	0 (0–1)	0.003
0–1, *n* (%)				
≥2, *n* (%)	93 (80.2)	44 (69.8)	49 (92.5)	0.002
	23 (19.8)	19 (30.2)	4 (7.5)	
Hospitalized in the previous year, *n* (%)	34 (29.3)	32 (50.8)	2 (3.8)	<0.001
Required systemic corticosteroids in the previous 6 months, *n* (%)	44 (37.9)	35 (55.6)	9 (17)	0.003
Number of corticosteroid cycles, median (IQR)				0
	0 (0–1)	1 (0–2)	0 (0–0)	
Treatment, *n* (%)				0.04
LAMA LAMA + LABA LABA + ICS LAMA + LABA + ICS	7 (6)	1 (1.6)	6 (11.5)	
	71 (61.2)	41 (67.2)	30 (57.7)	
	1 (0.9)	0	1 (1.9)	
	34 (29.3)	19 (31.1)	15 (28.8)	
Oxygen therapy, *n* (%)	47 (40.5)	32 (50.8)	15 (28.3)	0.014
Diagnostic procedures conducted for COPD evaluation				
COPD patients with serum vitamin D levels tested during follow-up	51 (44)	26 (41.3)	25 (47.2)	0.576
COPD patients with vitamin D deficiency who received vitamin D supplementation	33 (28.4)	12 (19)	21 (39.6)	0.021

Note: Data are presented as mean (SD), number (%), or median (IQR). Abbreviations: IQR: interquartile range; BMI: body mass index; mMRC: modified Medical Research Council; MET: metabolic equivalent of task; CAT: COPD Assessment Test; FEV1: forced expiratory volume in 1 s; FVC: forced vital capacity; GesEPOC: Spanish National Guideline for COPD; LABA: long-acting beta-2 agonists; LAMA: long-acting antimuscarinic agents; ICS: inhaled corticosteroids.

**Table 3 jcm-11-01347-t003:** Multivariate logistic regression to identify independent factors associated with presence of vitamin D deficiency.

Variable	OR (95% CI)	*p*
Activity level		
Medium-high (ref)		
Low	3.840 (1.541–9.567)	0.004
Inhaled corticosteroids		
No (ref)		
Yes	3.210 (1.230–8.317)	0.016
Number of systemic corticosteroid cycles		
No (ref)		
Yes	2.149 (1.330–3.473)	0.002

Note: OR: odds ratio.

## Data Availability

The data presented in this study are available on request from the corresponding author.

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
