# Peer review of "Testing for Vitamin D in High-Risk COPD in Outpatient Clinics in Spain: A Cross-Sectional Analysis of the VITADEPOC Study"

_jcm, 2022, doi:10.3390/jcm11051347_

Round 1

Reviewer 1 Report

Dear authors:

this is an interesting study for both pulmonologist and even more for GP. I only have a suggestion to be included in discussion and that is related to the  paragraph : This is surprising, since most patients with COPD in our study are at an increased 
risk of having low vitamin D blood levels due to the presence of risk factors. But it must be pointed out thas this is not true neither for chronic respiratory failure  (all patients had previous vit D determination) nor for osteoporosis (most patients have previous vit D measurement). So it should be mentioned that the other risk factors must also be taken into consideration. A paragraph dedicate to the low  Vit D levels in patients on inhaled corticosterois should also be added. 

Author Response

We appreciate this comment and we have added this paragraph in line 341:

Other risk factors associated with hypovitaminosis and which will be present in patients with chronic diseases like COPD should also be considered, such as: old age, institutionalized individuals, those with cognitive impairment, central obesity, smokers, corticosterois therapy and those with limited sun exposure.

Reviewer 2 Report

I found it to be an interesting work and with a large amount of data. But nevertheless, they could show some more graph about their results to make them more visual.

Yes, however, as I exposed in comments, having an abundant amount of data, my recommendation for the authors is, not only to organize the data in tables, but to include some more figure or graphical representation of them, in a way that facilitates the visualization of the results.

As for the language, I think it is correct. Finally, regarding the structure, I consider that the introduction, material and methods and discussion are adequate. However, I would improve the results with something more visual. The subject and technical aspects are correct.

Author Response

RESPONSE

We appreciate this comment and we have changed table 1 by a figure 1

Reviewer 3 Report

This is a high quality real life study on the prevalence of Vit. D Deficiency in people with COPD with far reaching practical consequences. It also highlights important clinical associations like corticosteroid use. The authors are also aware of the study limitations, so there are no major clinically relevant comments to make.

Author Response

We are grateful this comments.